# Predicting errors in accident hotspots and investigating satiotemporal, weather, and behavioral factors using interpretable machine learning: An analysis of telematics big data

Ali Golestani[1☉], Nazila Rezaei[1☉], Mohammad-Reza Malekpour[1], Naser Ahmadi[1,2], Seyed Mohammad-Navid Ataei [1], Sepehr Khosravi[1], Ayyoob Jafari[3], Saeid Shahraz[4], Farshad Farzadfar [1,2*]

1 Non-Communicable Diseases Research Center, Endocrinology and Metabolism Population Sciences Institute, Tehran University of Medical Sciences, Tehran, Iran, 2 Endocrinology and Metabolism Research Center, Endocrinology and Metabolism Clinical Sciences Institute, Tehran University of Medical Sciences, Tehran, Iran, 3 Faculty of Electrical, Biomedical and Mechatronics Engineering, Qazvin Branch, Islamic Azad University, Qazvin, Iran, 4 Institute for Clinical Research and Health Policy Studies, Tufts Medical Center, Boston, Massachusetts, United States of America

☉ These authors contributed equally to this work as the first authors.
* f-farzadfar@tums.ac.ir

## Abstract

### Background

Road traffic accidents (RTAs) are a major public health concern with significant health and economic burdens. Identifying high-risk areas and key contributing factors is essential for developing targeted interventions. While machine learning (ML) has been increasingly used to predict RTAs, the lack of interpretability limits its applicability in policymaking. This study aimed to utilize interpretable ML models to predict the occurrence of errors in road accident hotspots using telematics data in Iran and interpret the most influential predictors.

### Methods

We utilized data collected via telematics from 1673 intercity buses throughout the year 2020, spanning cities across all provinces of Iran. Merging this data with a weather-related dataset resulted in a comprehensive dataset containing location, time, weather, and error type variables. After preprocessing, 619,988 records without any missing values were used to train and compare the performance of six machine learning models including logistic regression, K-nearest neighbors, random forest, Extreme Gradient Boosting (XGBoost), Naïve Bayes, and support vector machine. The best model was selected for interpretation using SHAP (SHapley Additive exPlanation). Due to the high imbalance in the outcome, an ensemble approach was applied to train all models.

**Data availability statement:** The datasets generated and/or analysed during the current study are not publicly available due to the restrictions set by the funder of the study, National Institute for Medical Research Development (NIMAD). However, researchers with written permission can request to obtain the anonymized data. Requests to access the datasets should be directed to the NIMAD website (https://nimad.ac.ir/). A small sample of data is provided in the 'supporting information' section.

**Funding:** This work was supported by the National Institute for Medical Research Development (NIMAD), Tehran, Iran [grant number: 940567]. The funder had no role in study design, data collection and analysis, decision to publish, or preparation of the manuscript.

**Competing interests:** The authors have declared that no competing interests exist.

## Results

XGBoost demonstrated the best performance with an area under the curve (AUC) of 91.70% (95% uncertainty interval: 91.33% − 92.09%). SHAP values highlighted spatial-related variables, particularly the province of error and road type, as the most critical features for predicting errors in accident hotspots in Iran. Fatigue, as a behavioral error, was associated with a higher risk of predicting errors in accident hotspots, and certain weather-related variables including dew points and relative humidity also exhibited importance. However, temporal variables did not contribute significantly to the prediction.

## Conclusion

By integrating spatiotemporal, behavioral, and weather-related variables, our study highlighted the dominance of spatial factors in predicting errors in accident hotspots. These findings underscore the need for targeted road infrastructure improvements and data-driven policymaking to mitigate RTA risks.

## Introduction

Road Traffic Accidents (RTAs) represent a major global public health challenge, leading to significant mortality, morbidity, and economic burdens. The World Health Organization (WHO) estimated approximately 1.19 million annual fatalities due to road traffic injuries [1]. About 93% of road traffic deaths occurred in low- and middle-income countries (LMIC), despite having only 60% of vehicles [1]. In Iran, a lower-middle income country, RTAs rank as the second leading cause of Disability-Adjusted Life Years (DALYs) [2] and absorbs an estimated 2.19% of Iran's Gross Domestic Production (GDP) [3]. These statistics highlight the urgent need to investigate the determinants of RTAs to formulate effective policies and mitigate their impact.

Various internal and external factors can affect driving, potentially leading to aggressive driving, errors, and accidents [4]. According to Haddon's model, human, vehicle, and road-related factors—before, during, and after an accident—determine injury severity, with human factors contributing to 90%, vehicle-related factors to 30%, and road conditions to 10% of traffic accidents [5]. Notably, spatial variations in road traffic injuries demonstrate non-random cluster formations, indicating specific locations as more accident-prone like locations with higher traffic interactions and urban areas [6]. Furthermore, different road types influence driving styles and behavior [7]. Temporal factors, including seasons, weather conditions, hours, and days, all play crucial roles in the occurrence of RTAs [8]. An analysis of road traffic injuries in Iran revealed that most accidents occur during early evening/late afternoon and hours before noon [9]. Additionally, the highest incidence of accidents occurs during spring, summer, and early fall, with March marking the peak [9].

To better understand and predict driving errors leading to RTAs, researchers have employed various techniques incorporating factors such as vehicle speed,

acceleration patterns, braking intensity, steering movements, road conditions, and environmental factors like weather and traffic density [10,11]. Machine learning (ML) techniques, known for strong predictive performance, have gained widespread use in RTAs research [12]. Previous studies have shown good performance of ML models in identifying the accident hotspots, assessing accident severity, and classifying key contributing factors [13–15]. However, many ML models function as "black box" systems, lacking interpretability, which limits their application in policymaking [16]. Recent advancements in interpretable ML techniques address this challenge by maintaining predictive power while enhancing interpretability [17].

A promising source of data for ML-based RTA analysis is telematics—a technology that enables real-time vehicle data collection through telecommunications systems [18–20]. Telematics data includes details such as the location, speed, sharp turning, harsh acceleration/braking, the number of driving episodes, and distance traveled [21]. Previous studies have leveraged telematics in insurance and driver behavior analysis [22,23], demonstrating its value for risk assessment and prevention [24]. It has been used to classify drivers based on risky behaviors, allowing insurers to adjust premiums and implement risk mitigation strategies [24]. Additionally, telematics-based feedback has been shown to improve driver behavior and road safety [21,25]. While telematics data is widely utilized in the insurance industry, its potential application for road safety policymaking, particularly in LMICs, remains largely unexplored.

This study aimed to bridge this gap by integrating telematics data with accident hotspots data and weather-related variables data to predict errors occurrence in accident hotspots in Iran. We employed various ML models, ranging from simple and conventional to more advanced techniques, to develop the best-performing predictive model. Furthermore, we utilized interpretability techniques to identify the most influential factors contributing to errors occurrence in accident hotspots. The findings of this study provide valuable insights for policymakers, facilitating data-driven interventions and optimized resource allocation for accident prevention, particularly in resource-limited settings like Iran.

## Materials and methods

### Study design

In this retrospective study, we leveraged data collected from intercity buses that were initially enrolled in a trial study [26]. The primary focus of the original trial was to investigate the impact of non-punitive peer-comparison feedback on driving behavior, utilizing telematics devices [21,26]. The participants were all male bus drivers aged over 20. They were recruited between February 1, 2017, and August 31, 2017. A telematics device was installed on their vehicles, which included a total of 1,673 intercity buses traversing cities across all provinces of Iran. These devices were designed to remain operational post-trial and integrated into fleet management systems, ensuring the future use of telematics data for research. In this study, we used the dataset that included information collected over the full year of 2020, from January 1 to December 31. Fig 1 illustrates the flowchart of the steps taken in this study.

### Telematics device and data collection

Telematics is a system that integrates an onboard computer and a GPS to monitor vehicle behavior. The data collected from embedded modules (including GPS location, GPS speed, 3-axis acceleration, and date-time information) and the OBD port of the vehicle (covering diagnostic trouble codes, fuel consumption, and engine speed) is processed by a microcontroller. Subsequently, this information is transmitted every 10 seconds via a Global System for Mobile Communications (GSM) module to a GSM network, where it is recorded in a centralized data center [26]. After recording the precise location and capturing the latitude and longitude of each vehicle at 10-second intervals in the centralized data center, reverse geocoding through the Nominatim OpenStreetMap API was utilized to obtain the road type for each data point [26,27]. The considered road types included 'trunk', 'motorway', 'primary', 'secondary', 'tertiary', 'residential', and 'minor roads' with detailed definitions and distinctions provided elsewhere [28]. The specific errors were defined in accordance with Azmin

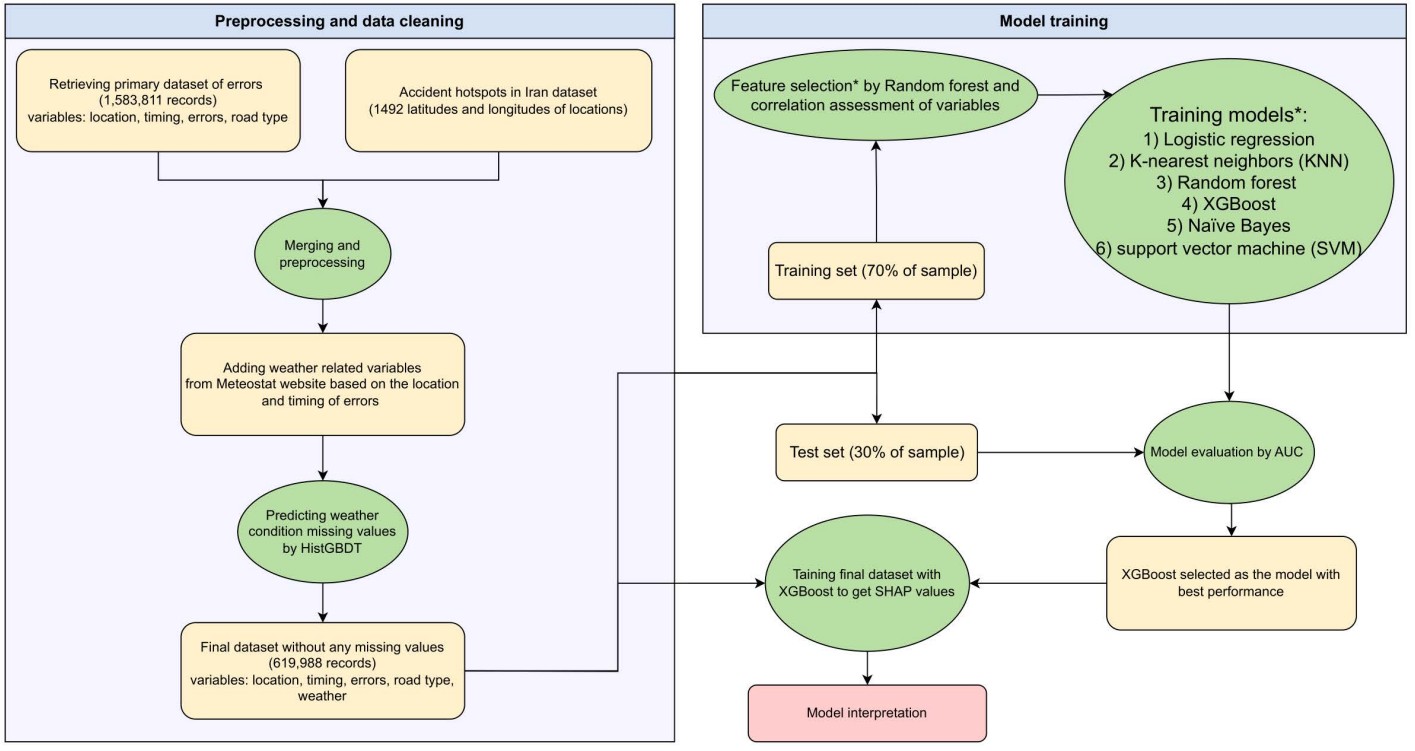

**Fig 1. The flowchart of this study steps.** *Models were trained in an ensemble approach to overcome outcome imbalance.

et al.'s [26] study as presented in Table 1. Instances in centralized data centers that were identified as errors based on predefined definitions were stored in a primary dataset. This primary dataset encompassing nearly 1.59 million records, made available for investigators in this study with information about the type of errors, the geographical location of errors (longitude and latitude), and the precise timing of errors (including date, hour, minute, and seconds), and road type. The distribution of the recorded errors by telematics in provinces of Iran is presented in Fig 2.

## Preprocessing and data cleaning

The primary dataset initially contained information solely about location, time, road type, and the type of error. To identify errors in road accident hotspots, we merged our dataset with one from the Iran Road Management Center (IRMC) including 1492 high-density accident locations by latitude and longitude [29]. The distribution of the hotspots in the provinces of Iran is presented in Fig 2 and S1 Table. These locations were determined using an index proposed by the Ministry of Roads and Urban Development of Iran [30]. Over the three-year study period (2018–2020), accident points within a 300-meter distance in a road with at least two fatal accidents or three accidents resulting in injury, or a combination of one fatal and two accidents resulting in injury, were considered as a location with potentially high-density accident occurrence. For each location, the accident index was calculated as follows:

$$\text{Accident index} = (1000 \times I)/(365 \times A \times T)$$

where T is the number of study years (three in this study), A is a coefficient based on road type (higher for main roads), and I is related to the severity of accidents, calculated using the formula:

**Table 1. Definition of error types using telematics devices in this study.**

| Error type | Definition |
| --- | --- |
| **Harsh braking (deceleration)** | x-axis deceleration lesser than −0.4 g(km.h⁻¹s⁻¹) [26] |
| **Harsh acceleration** | x-axis acceleration more than 0.22g (7.8 km.h⁻¹s⁻¹) [26] |
| **Harsh turning** | y-axis acceleration more than 0.7g (24.7 km.h⁻¹s⁻¹) [26] |
| **Over speed** | Violation occurs when the vehicle speed exceeds the speed limit by more than 20%, determined by comparing the vehicle speed sent by telematics with the speed limit of the vehicle location. Speed limit information is obtained from Nominatim OpenStreetMap API, considering road geography, type, and speed limits [26,27]. |
| **Fatigue** | Continuous driving for more than 240 minutes without a 20-minute rest [26] |

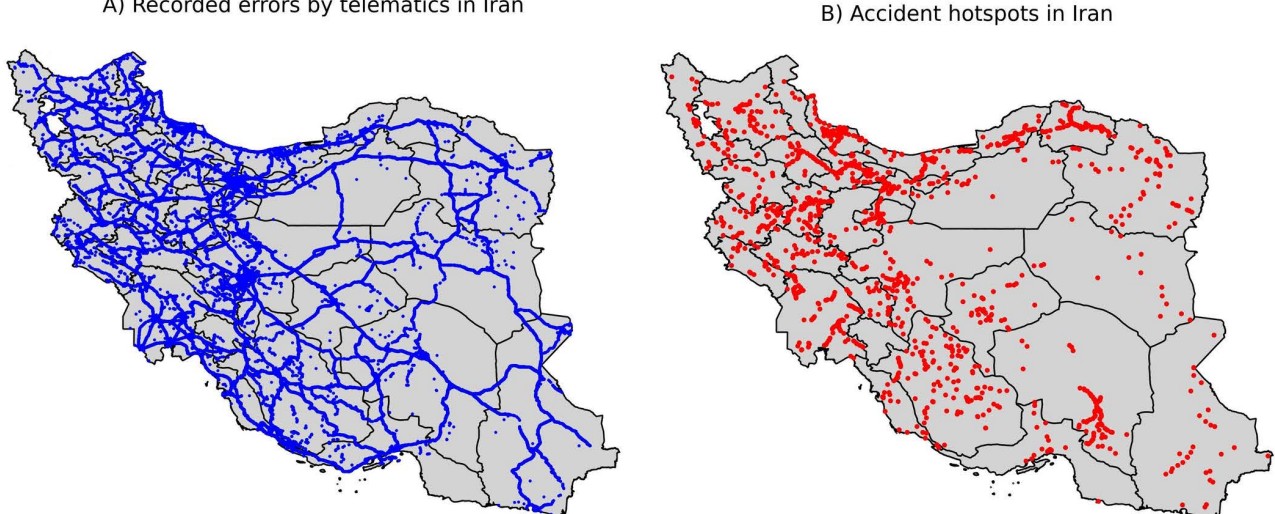

**Fig 2. The distribution of errors recorded by telematics and accident hotspots in provinces of Iran.** A) Errors recorded by telematics. B) accident hotspots.

$$l = x + 3y + 9z$$

where x is the number of accidents causing vehicle damage without injury or death, y is the number of accidents resulting in injury, and z is the number of fatal accidents. Locations with an index value above the mean were considered high-density accident locations, or accident hotspots. To determine if an error from our telematics dataset occurred in an accident hotspot, we employed the Haversine formula [31] to calculate the distance between each error location and any of the locations in the hotspot dataset. If any distance was less than 150 meters, we considered the error to have occurred in the accident hotspot. We utilized recorded locations to identify the province in which the error occurred. Subsequently, we used the time of the errors to create columns related to the season, month, day, and hour. Additionally, we categorized each day based on whether it was a workday, holiday, long holiday (more than one subsequent holiday day), or a pre-long holiday day. The exact time of the error was also used to determine the ambient light situation, including day, night, or twilight, based on the time of sunrise and sunset on that specific date.

To incorporate weather-related variables, we sent requests to the Meteostat website API [32], including the exact location and time of the error using the Scrapy package [33]. However, it's important to note that a significant portion of the requested

weather data was unavailable. Only a subset of the dataset, comprising approximately 690 thousand records, contained at least one non-missing weather-related variable. These variables included temperature (in Celsius (°C)), dew point (in °C), relative humidity (in percentage), wind direction (in degrees), average wind speed (in kilometers per hour), sea-level air pressure (in hectopascal (hPa)), and weather conditions (classified as clear, cloudy, foggy, rainy, snowy, or stormy). Among these variables, weather condition non-missing values comprised about 300 thousand rows of the dataset, roughly half of the other weather-related variables. To maximize data utilization, we used latitude, longitude, month, ambient light situation, hour, and other weather-related variables of each error to impute missing values in the weather conditions. For this purpose, 70% of dataset with non-missing values of weather conditions variable, was used to train histogram-based gradient boosting decision trees (HistGBDT) due to its ability to handle missing values in features [34–36] The performance was optimized by grid search, and was evaluated with the remaining 30% of the dataset that showed an accuracy of 93% and a weighted average F1-score of 92%. This model was then used to impute the weather condition for all instances with missing values for this variable. Due to the large dataset and the study's objective of assessing all possible variables, we restricted our ML analysis to data without any missing values, resulting in a final dataset comprising 619,988 rows.

To mitigate the impact of variations in quantitative variables units across different models in the next steps, all quantitative features were standardized using the 'StandardScaler' from the Scikit-Learn package [35]. Additionally, if it was required, all categorical variables were one-hot encoded before modelling.

## Outcome imbalance

During the initial exploratory analysis of our dataset, we identified a significant imbalance in the outcome variable, which represented errors in road accident hotspots. Specifically, the prevalence of errors in road accident hotspots was only about 1.9% in the dataset.

This imbalance could result in falsely high accuracy, as the model might simply assign all outcomes to the majority class, failing to effectively capture the minority class. To mitigate this issue, various techniques could be employed. Instead of opting for conventional methods such as over-sampling (e.g., synthetic minority oversampling (SMOTE)) or under-sampling (e.g., K-nearest neighbors (KNN models)), we chose to leverage ensemble models [37]. In ensemble models, the minority class remains fixed, and in each iteration, a random sample is drawn from the majority class for training. This process is repeated over several iterations, allowing the model to train on different subsets of the main dataset. The final decision of the model for a test dataset is determined by the aggregated vote of all trained models, similar to other ensemble models. Throughout the steps of feature selection, grid search, and comparing the best-performing models, we utilized ensemble models with varying numbers of estimators based on the model type and computational efficiency. Specifically, we employed the BalancedRandomForestClassifier [38], EasyEnsembleClassifier [39], and BalancedBaggingClassifier [40] for random forest (RF), Extreme Gradient Boosting (XGBoost), and other models, respectively, from the 'imblearn' library [37].

## Train/test splitting of dataset

We split the dataset into 70% training and 30% testing sets. We kept the test set separate from the training set at all stages, including feature selection and model development, and used it only for evaluating the performance of each model. This approach ensures the training set remains uncontaminated by any information from the test set, thereby ensuring the validity of model evaluations.

## Feature selection

To identify the most influential predictors for our models, ensuring optimal performance and reduced model complexity, we employed feature selection techniques involving balanced random forest and the examination of variable correlations. For feature selection, a balanced random forest classifier with 1000 estimators, with replacement, and a maximum depth of 100, was trained on the training set, using the occurrence of errors in accident hotspots as the outcome. The subsequent

model was utilized for permutation importance analysis on the test dataset, aiming to discern the significance of each predictor. Permutation feature importance quantifies the reduction in a model's score when the values of a single predictor are randomly shuffled [41]. In this study, permutation analysis was performed on 1000 random samples, each with the same size as the minority outcome, obtained through bootstrapping from the majority outcome data. This process was iterated 10 times. This method was chosen for its robustness, particularly in handling the high cardinality of categorical variables, such as province and hour in our dataset [41]. Additionally, we calculated various correlation metrics—Pearson correlation, Cramér's V, Point Biserial Correlation, and Correlation Ratio—to assess relationships among quantitative-quantitative, categorical-categorical, quantitative-binary categorical, and quantitative-non-binary categorical variables, respectively. Predictors with no negative permutation importance scores were selected for further analysis. In instances of high correlation among selected predictors, we prioritized the predictor with a higher permutation importance score.

## Machine learning models and evaluation

To assess the predictive performance on our dataset, we compared the outcomes of six distinct models: (1) logistic regression [16], (2) K-nearest neighbors (KNN) [42], (3) random forest (RF) [42], (4) Extreme Gradient Boosting (XGBoost) [43], (5) Naïve Bayes [42], and (6) support vector machine (SVM) [42]. Notably, a linear SVM was chosen to accommodate the large size of our dataset, a factor limiting computational capacity when using SVM [44]. Models were trained on the training dataset. While various approaches such as grid search, random search, and Bayesian optimization exist for model optimization, grid search has demonstrated comparable performance [45]. Therefore, it was selected for hyperparameter tuning in this study. Grid search was conducted using stratified 5-fold cross-validation with 3 repeats. Performance metrics were calculated during hyperparameter tuning, with the selection of the best-performing model determined using the receiver operating characteristic (ROC) plot area under the curve (AUC) metric. After identifying the best-performing model for each of the six machine learning algorithms, performance evaluation across different models was conducted by comparing the 95% uncertainty interval (UI) of AUC, obtained by bootstrapping with 1000 iterations on the test dataset. Considering other performance evaluation metrics, AUC was specifically chosen for its robustness in the presence of outcome imbalance. The models were implemented using the Scikit-Learn and XGBoost packages [36,46].

## Model interpretation

After selecting the best-performing model, we employed SHapley Additive exPlanations (SHAP) to interpret the model [47]. SHAP is a novel game-theory-based approach that calculates the contribution of each feature to the model's predictions [17]. It assigns an importance value (SHAP value) to each feature, explaining the role of each feature in predicting the outcome for each observation. These importance values can be summarized using visualizations such as bee swarm plots or the mean of the absolute SHAP values for each feature. Consider a model where a set $N$ with n features is utilized to predict an output $v(N)$. In SHAP, each feature's contribution ($\phi_i$ representing the contribution of feature $i$) on the model output $v(N)$ is calculated based on their marginal impact [48]. Using axioms that guarantee a fair distribution of each feature's contribution, SHAP values are determined as follows:

$$\phi_i = \sum_{S \subseteq N[i]} \frac{|S|!\,(n-|S|-1)!}{n!} \left[v\left(S \cup \{i\}\right) - v(S)\right]$$

An additive feature attribution method defines a linear function for binary features $g$ as follows:

$$g\left(z'\right) = \phi_i + \sum_{i=1}^{M} \phi_i z'_i$$

where $z' \epsilon \ [0, 1]^M$ represents 1 if a feature is present and 0 if it is not, and $M$ denotes the total number of input features [49]. Given the additive properties of SHAP values, we initially computed the SHAP values for each predictor for every 619,988 instances in the dataset using our ensemble model. Subsequently, for each predictor, we calculated the mean of all SHAP values calculated for each instance in different iterations, consolidating them as the final SHAP values for that specific observation. As our outcome variable was categorical, and considering the values of each feature, higher feature values (or their presence in the case of categorical features) aligning with higher SHAP values indicated an association with a higher prediction of errors occurring in hotspots, and vice versa. Conversely, lower feature values (or their absence in the case of categorical features) aligning with higher SHAP values showed an inverse association of the feature with the prediction of errors occurring in accident hotspots, and vice versa. The 'SHAP' package was utilized for computing SHAP values and visualizing the results [50]. We also considered performing subgroup analysis if certain variables showed significant patterns.

### Statistical analyses

Quantitative variables were described as the mean and standard deviation (SD), while qualitative variables were presented in the form of frequencies and percentages. A 95% confidence interval (95% CI) was utilized to indicate the uncertainty of an estimation and evaluate statistical significance. All processes related to data cleaning, modeling, and visualizations were executed using Python programming software version 3.10.13 (https://www.python.org/). A small sample of the data and the codes used for analysis are provided in S1 and S2 Files, respectively.

### Ethical consideration

This study performed according to the Declaration of Helsinki. Before the installation of the device, all participating drivers provided written informed consent, which included a primary explanation of the project. They were informed their real-time driving data would be gathered and used in subsequent projects. Participants were assured that their data would remain anonymous and confidential. This project has received ethical approval from the National Institute for Medical Research Development (NIMAD), Tehran, Iran (ethics code: IR.NIMAD.REC.1394.016).

## Results

### Overview

After the initial cleaning of the primary telematics dataset, a total of 1,583,811 errors were available for analysis. Among these records, only 29,618 (1.87% [95% CI: 1.85–1.89]) occurred in identified accident hotspots. Examining error types revealed that the majority of errors were harsh turning (57.02% [56.95–57.10]) and fatigue (29.51% [29.44–29.58]). Regarding road types, trunks (51.99% [51.91–52.07]) and primary roads (21.01% [20.95–21.07]) accounted for the majority of errors. There were notable temporal patterns, with most errors occurring during the time intervals of 2–7 and 18–20 throughout the day. Surprisingly, the occurrence of errors was significantly lower in spring (19.08% [19.02–19.14]) compared to other seasons. January stood out as the month with the highest number of errors (11.21% [11.16–11.26]). Errors were predominantly recorded during daylight hours (47.61% [47.53–47.69]) and were more frequent on workdays (67.14% [67.07–67.22]), with Sunday, a workday in Iran, exhibiting the highest error rate at 14.94% (14.88–14.99). Analyzing errors across provinces highlighted Isfahan (13.09% [13.04–13.15]), Tehran (12.97% [12.92–13.02]), and Bushehr (12.14% [12.09–12.19]) as containing the highest error rates. Interestingly, the proportion of errors in hotspots was notably higher in Golestan (10.52% [10.00–11.05]), Hamadan (6.61% [6.35–6.86]), and Kurdistan (6.46% [6.24–6.68]) compared to other provinces. Additional details on the descriptive summary of variables in the dataset are provided in S2 Table. A descriptive summary of weather-related variables and the corresponding number of records with non-missing values are presented in Table 2.

## Feature selection

A descriptive summary of records without missing values after imputing weather condition variable is provided in S3 Table. Evaluation of the trained balanced random forest classifier for feature selection on the test set demonstrated a weighted average F1-score of 89.00% and an AUC score of 90.90%. The results of permutation importance, depicted in Fig 3, revealed that province, road type, and error type exhibited notably higher importance compared to other features. Dew point, weather condition, hour, average wind speed, and relative humidity also showed positive permutation importance values. Furthermore, we assessed the correlation among different variables, presenting the heatmaps in S1 Fig. The analysis indicated that the variables selected by permutation importance were not strongly correlated with each other. Consequently, all eight selected variables were considered for model training in the subsequent steps.

## Model evaluation

The hyperparameters tuned during grid search, along with those of the best-performing model, are detailed in S4 and S5 Tables. Among the best-performing models of each type, XGBoost emerged as the most performant, achieving an AUC of 91.70% (95% UI: 91.33 − 92.09%). Notably, RF (AUC: 91.14% [95% UI: 90.72 − 91.55%]) and KNN (90.09% [89.60 − 90.58%]) demonstrated competitive performance. Conversely, SVM exhibited the least favorable performance among the trained models, by an AUC of 82.03% (81.37 − 82.66), inferior to both Naïve Bayes (84.77% [84.09 − 85.40%]) and Logistic Regression (85.73% [85.15 − 86.28%]) (Fig 4). Consequently, the XGBoost model, with its superior AUC and balanced accuracy of 84.70%, was selected for interpretation. Additional performance evaluation metrics of models are detailed in S6 Table.

## Model interpretation

Fig 5, which illustrates the mean absolute SHAP values for features exceeding 0.05, underscores the pivotal impact of location-related, road-type, and error-type features on the model's predictive decisions. Tehran province significantly influenced model decisions (mean absolute SHAP = 0.74), and trunk roads (0.36), fatigue (0.33), and Hamadan province (0.32) also played substantial roles. Among weather-related variables, dew point emerged as a prominent contributor, boasting the highest mean absolute SHAP value (0.14). S2 Fig provides a comprehensive view of the mean absolute SHAP values for all features.

Fig 6 delineates the impact of features with a mean absolute SHAP greater than 0.05 on the model output, while SHAP values for all features are presented in S3 Fig. Error occurrences in Tehran were associated with negative SHAP values, indicating a connection to non-hotspot accidents. Conversely, errors in Hamadan, Bushehr, and Kurdistan showed higher SHAP values, signifying associations with accident hotspots. For road types, errors in trunks were primarily linked to accidents in non-hotspots, while errors in motorways exhibited positive SHAP values, indicating occurrences in accident hotspots. Fatigue exhibited a remarkable association with higher SHAP values, signifying a strong link with accidents in hotspots. Among other error types, harsh turning did not positively contribute to the prediction of occurring errors in the accident hotspot. Weather-related variables showed diverse SHAP values based on their values. Higher dew points and lower relative humidity were especially related to lower SHAP values, indicating errors in non-hotspots. Foggy weather was associated with lower SHAP values, while cloudy weather was predominantly associated with higher values. Other weather conditions represented modest SHAP values. Hour 19 showed the highest SHAP values, indicating a significant association with occurrences in hotspots.

Due to the significance of the provinces in the analyses, we conducted subgroup analyses. In these subgroup analyses, we repeated analysis on the data from three important provinces including Tehran, Hamadan, and Bushehr. The results are shown in S4 Fig. Different variables showed importance in each province. While in Tehran, fatigue was

**Table 2. Descriptive summary of weather-related variables in the primary dataset.**

**Categorical variables**

| Variable | Error occurrence in accident hotspots (Number, percentage, 95% confidence interval) | | Total (Number, percentage, 95% confidence interval) |
|---|---|---|---|
| | No | Yes | |
| **Weather condition** | | | |
| Clear | 188000 (98.14%)(98.08-98.2) | 3570 (1.86%)(1.8-1.92) | 191570 (68.05%)(67.87-68.22) |
| Cloudy | 28453 (98.37%)(98.23-98.52) | 470 (1.63%)(1.48-1.77) | 28923 (10.27%)(10.16-10.39) |
| Foggy | 41904 (99.16%)(99.07-99.24) | 357 (0.84%)(0.76-0.93) | 42261 (15.01%)(14.88-15.14) |
| Rainy | 13466 (98.37%)(98.16-98.58) | 223 (1.63%)(1.42-1.84) | 13689 (4.86%)(4.78-4.94) |
| Snowy | 2377 (97.42%)(96.79-98.05) | 63 (2.58%)(1.95-3.21) | 2440 (0.87%)(0.83-0.9) |
| Storm | 2605 (98.45%)(97.98-98.92) | 41 (1.55%)(1.08-2.02) | 2646 (0.94%)(0.9-0.98) |
| Total | 276805 (98.32%)(98.27-98.37) | 4724 (1.68%)(1.63-1.73) | 281529 (100.0%)(100.0-100.0) |
| Missing values | 1277388 | 24894 | 1302282 |

**Quantitative variables**

| Variables (unit) | Error occurrence in accident hotspots (Mean±standard deviation) (proportion in non-missing) | | Total (Mean±standard deviation) (number of non-missing) |
|---|---|---|---|
| | No | Yes | |
| Temperature (°C) | 19.38±11.79 (98.42%) | 17.38±12.13 (1.58%) | 19.35±11.80 (671654) |
| Dew point (°C) | 4.54±9.57 (98.42%) | 3.99±8.09 (1.58%) | 4.53±9.55 (671523) |
| relative humidity (%) | 46.11±25.72 (98.42%) | 49.47±25.90 (1.58%) | 46.16+25.73 (671251) |
| wind direction (degrees) | 162.60±116.92 (98.44%) | 157.51±113.60 (1.56%) | 162.52±116.87 (663861) |
| average wind speed (kilometers per hour) | 10.09±7.55 (98.38%) | 9.20±7.47 (1.62%) | 10.08±7.55 (637718) |
| sea-level air pressure (hPa) | 1013.95±8.57 (98.43%) | 1014.19±7.98 (1.57%) | 1013.96±8.55 (61598) |

°C: Celsius, hPa: hectopascal

associated with errors in accident hotspots, in Hamadan and Bushehr, harsh turning and primary roads were the most important variables.

## Discussion

Utilizing a telematics system to aggregate data on predefined errors within intercity buses across all provinces of Iran and integrating this information with weather data, our study explored various machine learning models to predict the occurrence of errors in accident hotspots. Ultimately, we achieved optimal predictive performance with an XGBoost model with an AUC of 91.70%, which is considered as an outstanding performance based on AUC interpretation [51]. Furthermore, our findings underscored the location of errors (province of occurrence) and road types as the most important predictors in anticipating errors in accident hotspots. Additionally, behavioral factors, specifically driver fatigue, emerged as a crucial predictor, while variables associated with weather and time exhibited the least contribution to predicting errors in hotspots. Thus, this study indicated that factors related to the location were more important in predicting the occurrence of errors in accident hotspots compared to other factors.

Our study showed that decision tree-based models, including XGBoost and Random Forest, had the best performance. Similarly, other studies have also demonstrated strong performance of decision tree-based models [52,53]. A comprehensive examination of a 6-year dataset from Michigan Traffic Agencies (MTA) with nearly 270 thousand crashes, revealed that RF models could accurately predict injury severity with an accuracy of 75.5%. This study utilized variables related to driver demographics, environmental factors, and behavior [13]. Another study in Portugal identified RF as the

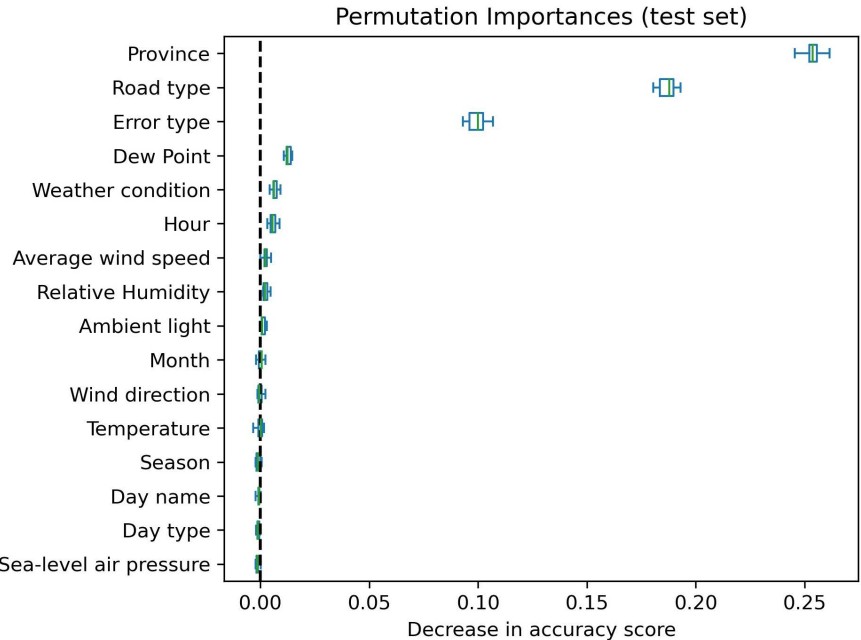

**Fig 3. Feature selection using permutation importance on test dataset of the balanced random forest model.**

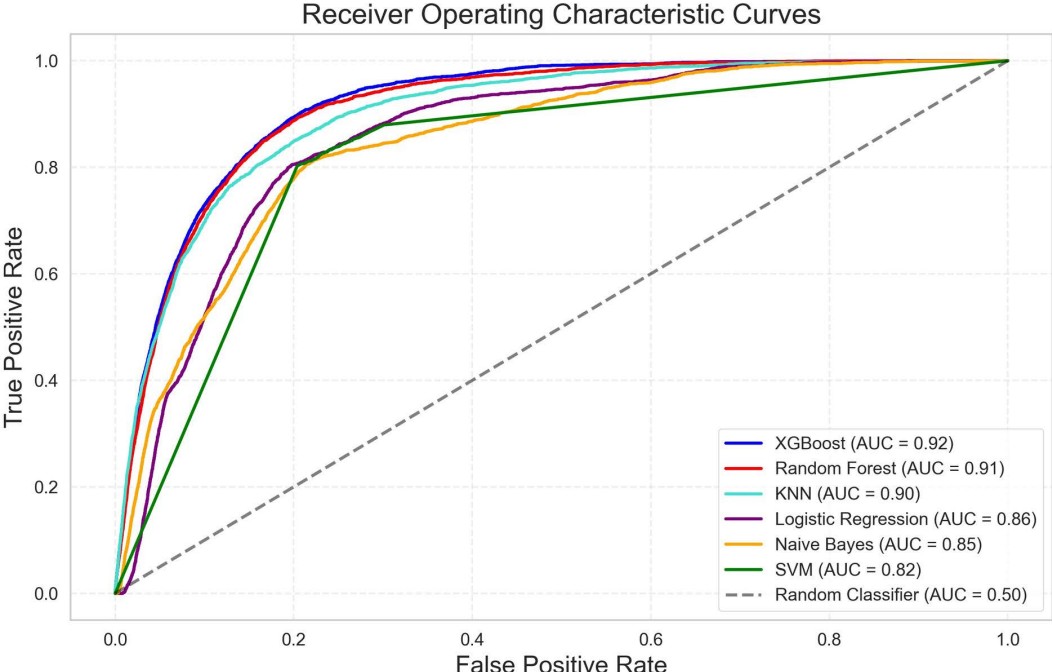

**Fig 4. Receiver operating characteristic (ROC) curve for prediction of error occurrence in accident hotspots for all evaluated ML models with their corresponding AUC.** ML: machine learning, AUC: area under the curve.

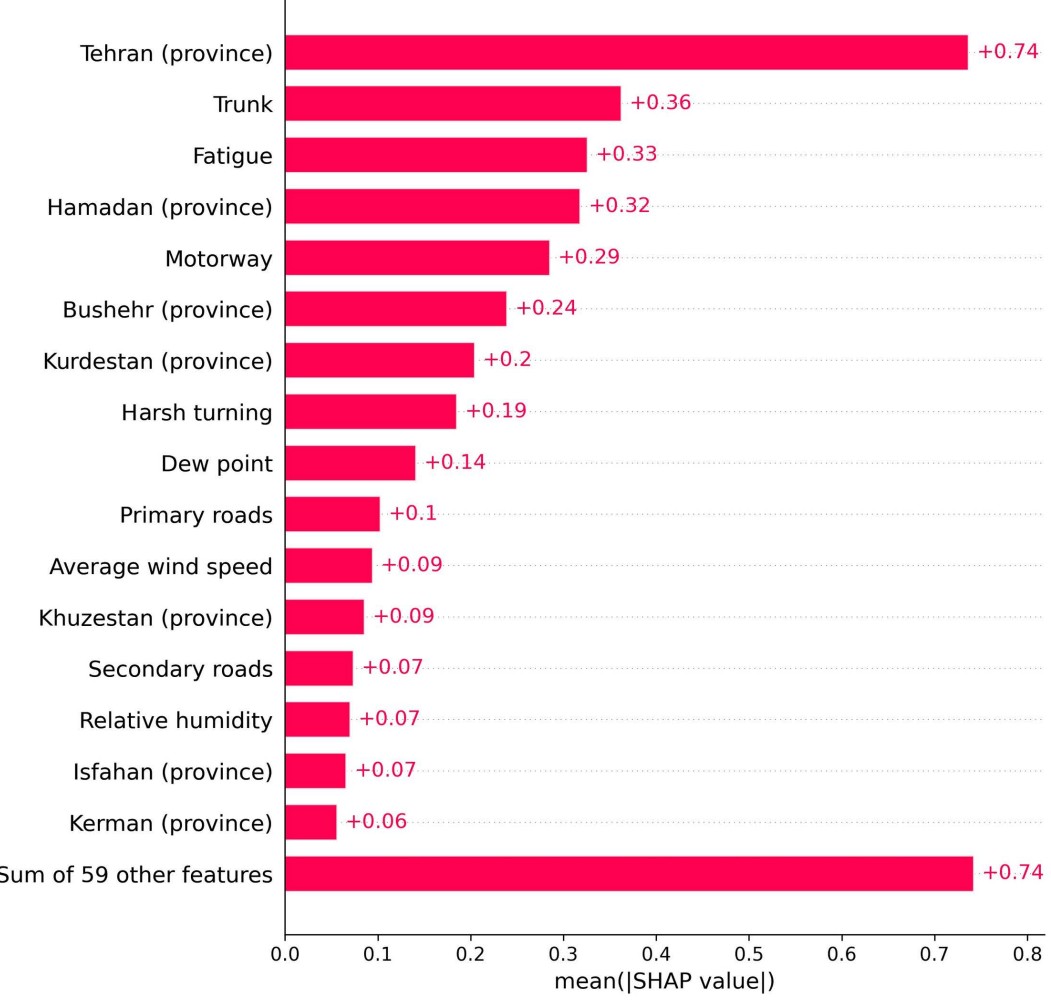

**Fig 5. The mean absolute SHAP values for predictors with values exceeding 0.05.**

best-performing model for predicting accident hotspots, achieving an AUC of 0.68 and an accuracy of 73% [14]. In the analysis of Nebraska crash data, KNN exhibited the most effective performance in predicting accident severity, followed by RF [54]. Another study, exploring various machine learning algorithms for road accident prediction, identified a model achieving a 61% prediction rate with a false alarm rate of 38% [55]. In another study, the RF outperformed other models in predicting injury severity, achieving higher accuracy in individual classes as well as overall prediction performance [56]. An alternative study, aimed at detecting accident occurrences using real-world data from Chicago metropolitan expressways, employed the XGBoost model, achieving an AUC of 89% and near-perfect accuracy [48]. Notably, their dataset also suffered from imbalance, yet the authors addressed this issue using SMOTE. While prior research has shown XGBoost's superiority over neural networks in the context of balanced structured datasets, and its effectiveness in handling the complex data distribution within the feature space [57], both our study and the work by Parsa et al. [48] demonstrate with the appropriate approach to resolving data imbalance and subtle feature selection, XGBoost—a newer boosting decision tree introduced in 2016 [43]—can be an ideal model for predicting events. Its computational efficiency and ability to handle missing values have contributed to its growing popularity in recent years [58]. While XGBoost is often claimed to possess

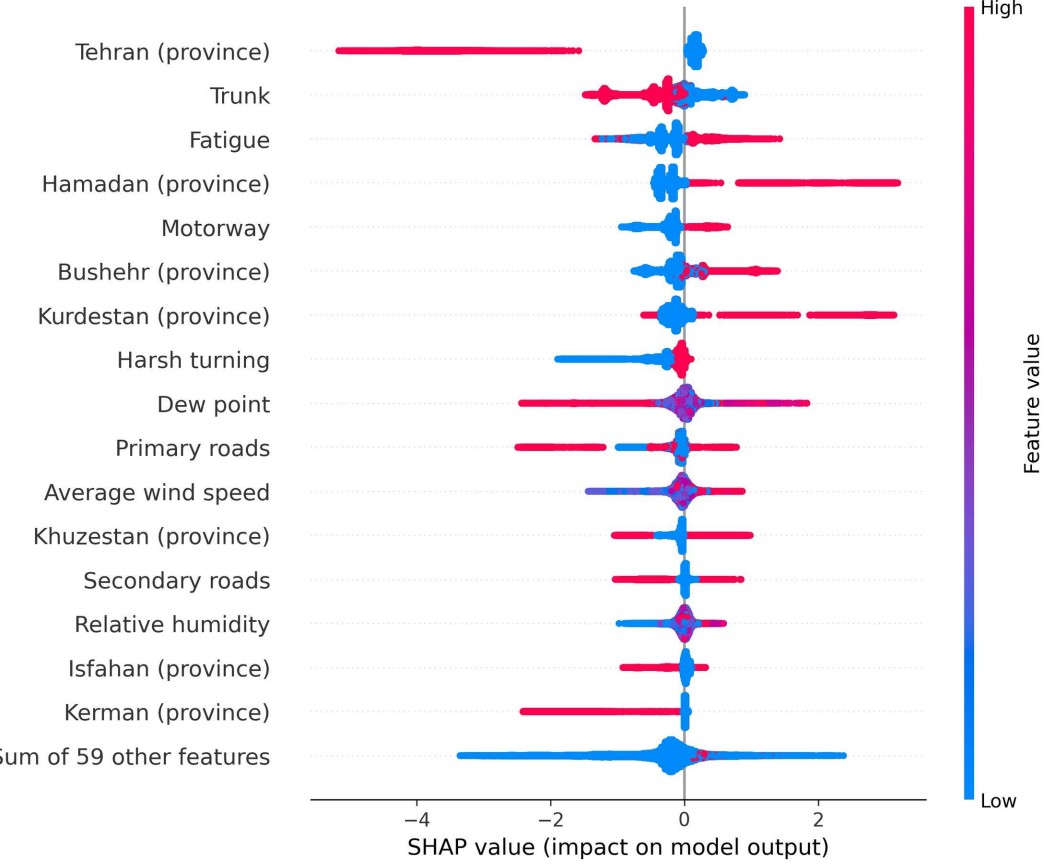

**Fig 6. SHAP beeswarm summary plot of the most important predictors for error occurrence in accident hotspots.**

superior predictive performance, concerns have been raised regarding its applicability to tasks requiring interpretability [59]. However, the use of SHAP, a tool that overcomes the limitations associated with similar tools, has become prevalent [58]. This combination of XGBoost and SHAP has demonstrated effectiveness in achieving both predictive accuracy and interpretability in similar studies [60–63]. This makes it a strong option for future research seeking both high accuracy and explainability.

In our study, feature selection based on permutation importance and model interpretation highlighted the significance of some previously evaluated predictors in forecasting errors in accident hotspots. The analysis revealed variables related to the time of the error had lesser importance. This contrasts with other studies on accident occurrence in Iran, which have demonstrated a notable association between time and accidents [9]. A study reported that approximately 28% of all accidents and about one-third of fatal accidents occurred in summer, particularly in August [64]. Regarding the occurrence of accidents on different days, Friday had the fewest accidents, and most road traffic injuries occurred on the day preceding long holidays, possibly due to heavier traffic on these days [64]. Data collected from the Iranian Legal Medicine Organization indicated a fluctuating injury rate throughout the year, following a periodic pattern with peaks in spring and summer, and declines in autumn and winter [65], consistent with injury death patterns [66]. However, it is essential to note that these differences could stem from methodological variances and the specific focus of each study. While time-related variables are potentially associated with the occurrence of accidents, they did not hold the same level of importance in predicting errors in hotspots when considered alongside other predictors in our model.

Our study highlighted the significance of spatial-related variables, particularly the province of error and road type, as the most important features for predicting errors in accident hotspots in Iran. Our findings revealed that the occurrence of accidents in Tehran, the capital province of Iran, is associated with a lower prediction of hotspot accidents, while provinces like Hamadan, Bushehr, and Kurdistan show a higher risk of error occurrence in hotspots. This aligns with previous studies that demonstrate certain clusters are more prone to accidents. Clustering methodologies have been employed in studies identifying accident hotspot zones and their spatial distribution in London, leading to the conclusion that location-specific policies should be adapted [67]. Another study utilized clustering methods to identify accident hotspots and then evaluated the impact of environmental road structure on drivers' behavior [68]. While approximately 75% of accidents happen in city streets, fatal accidents are more common in rural areas, possibly due to a higher likelihood of dangerous behaviors such as overspeeding on outer city roads [64]. Factors related to road types, such as lane width, median existence, median type, shoulder width, horizontal curves, and vertical curves, can influence driving style and drivers' behavior [69]. Additionally, regulations such as speed limits, the presence of cameras, traffic signs, and features like congestion levels can vary by road type [70]. While it is known that location-related variables are important, our study showed they were more significant than behavioral and weather variables. We identified specific provinces and road types associated with higher or lower error risks, guiding policymakers on resource allocation. Our findings suggest that improving location-related conditions in Iran can significantly reduce errors in hotspots. The subgroup analysis of the three most important provinces revealed that different variables are important in each province, highlighting the necessity of province-specific strategies and the evaluation of each province situation to avoid generalizing risk variables. Our study interpreted fatigue as one of the behavioral errors associated with a higher risk of predicting accident hotspots, and certain weather-related variables also demonstrated importance in hotspot prediction. The relationship between a fatigued or sleepy state and a decline in performance has been well-established, leading to increased accident risks and errors [71]. Thus, detecting driver fatigue with simple strategies like assessing the continuity of driving time would reduce accident risks. These strategies could operate at both individual and governance levels, such as sending alerts to drivers who have been driving for extended periods or establishing penalties for prolonged driving. Additionally, drivers' behavior is influenced by weather conditions due to their impact on speed, road surfaces, and visibility [69,70]. Therefore, utilizing weather data would be beneficial for predicting potential driver behaviors in each location.

While the use of telematics in Iran and similar countries is not yet common, exploring its potential applications could lead to its wider integration in vehicles. It has been shown machine learning techniques are effective in providing alert information to travelers in unfamiliar locations, thereby reducing accidents [72]. Prior telematics studies have demonstrated its cost-effectiveness in monitoring and changing driving performance in Iran [21,26,73]. While informational interventions alone may not be sufficient, combining them with immediate consequences, such as prompt punishments, has proven effective in altering driver behavior [74]. Considering these insights, telematics emerges as a cost-effective option for vehicle and insurance companies. By aggregating real-time vehicle location data, integrating it with other relevant data sources, including weather and environmental variables, and identifying predefined errors such as drivers' fatigue or risky driving behaviors, telematics systems can send alert messages to drivers at high risk of making errors in accident hotspots.

## Strengths and limitations

Our study had several limitations that should be considered. First, our analysis focused on the behavioral errors of bus drivers—a group with specialized driving skills. Extrapolating these findings to the general population requires caution, as professional drivers have unique characteristics that may not fully represent all road users. Second, bus drivers typically operate on fixed and familiar routes, which may influence the nature of their errors compared to drivers navigating unfamiliar roads. Additionally, the findings of this study may not be directly applicable to other regions or transport systems due to differences in infrastructure, regulations, and driving behaviors. Another limitation is the study's

reliance on data from a single year. Without longitudinal data spanning multiple years, it is difficult to assess trends, seasonality, or long-term changes in predictor variables. Future research could incorporate multi-year datasets to predict the frequency of error occurrences in accident hotspots, providing policymakers with more reliable insights for road safety interventions [75]. Moreover, to streamline the model, we used a limited set of variables related to behavioral errors. While these factors are crucial, other variables—such as personality traits, age, and education level—may also influence driving behaviors. Research suggests that dangerous driving behaviors are less prevalent among men, individuals with intermediate education levels, and the elderly [76]. Additionally, factors such as road maintenance, law enforcement measures, and socio-economic conditions could further impact error risk at accident hotspots. Future studies should incorporate a broader range of driver-, vehicle-, and environment-related variables to develop more comprehensive predictive models. Although we implemented thorough data preprocessing, potential errors or biases in telematics and weather data collection remain. The study assumes that all variables were accurately measured and recorded. Future research should utilize datasets with a wider range of validated variables to improve analytical depth and accuracy.

Despite these limitations, our study has several strengths. First, data collection via telematics devices reduces information bias compared to self-reported driver data. The integration of large-scale data from multiple sources enhances the robustness and interpretability of our findings. By incorporating spatial, temporal, behavioral (fatigue), and weather-related variables, our study provides a multifaceted perspective on road safety and highlights the interactions between key predictive factors. Furthermore, we employed six different machine learning models, ranging from simple to advanced techniques, ensuring a comprehensive evaluation of predictive performance rather than relying on a single algorithm. The use of SHAP values enhances the interpretability of our models by identifying the most influential predictors, facilitating actionable policy recommendations. Finally, we addressed the significant imbalance in the outcome variable by applying an ensemble approach, which improves predictive reliability and reduces bias against the minority class.

## Conclusion

In conclusion, this study employed machine learning techniques to develop a predictive model for identifying occurrences of errors in road accident hotspots, utilizing a telematics dataset integrated with weather information in Iran, a low-middle-income country. The XGBoost model, achieving an AUC of 91.70%, demonstrated outstanding predictive performance. While it was known location-related variables were important, SHAP values showed that province and road type were the most important predictors among different spatiotemporal, weather, and behavioral factors. This finding suggests a need for prioritized attention from policymakers toward these two factors and the development of province-specific strategies. For future research, employing larger datasets with additional variables can contribute to the development of more accurate and reliable predictive models.

## Supporting information

**S1 Fig. Evaluating correlation among variables using different correlation techniques.** A) Pearson's correlation for quantitative variables, B) Cramér's V for categorical variables, C) Correlation of categorical vs quantitative variables. (PNG)

**S2 Fig. The mean absolute SHAP values for all predictors.** (PNG)

**S3 Fig. SHAP beeswarm summary plot of all predictors.** (PNG)

**S1 Table. Distribution of accident hotspots in Iran.**
(DOCX)

**S2 Table. Descriptive summary of variables excluding weather-related variables in the primary dataset.**
(DOCX)

**S3 Table. Descriptive summary of variables in the dataset after prediction of weather condition and without any missing values.**
(DOCX)

**S4 Table. Models hyperparameters tuning.**
(DOCX)

**S5 Table. Performance of different models during hyperparameters tuning by grid search.**
(XLSX)

**S6 Table. Evaluation of machine learning models for the prediction of error occurrence in accident hotspots.**
(DOCX)

**S1 File. A small sample of data used for analysis (in pickle format).**
(PKL)

**S2 File. Codes used during analysis process.**
(IPYNB)

## Acknowledgments

The authors gratefully acknowledge the National Institute for Medical Research Development (NIMAD), Tehran, Iran for its support. We also thank Kavi Bhalla, PhD, for his contribution to the revision of the manuscript.

## Author contributions

**Conceptualization:** Ali Golestani.

**Data curation:** Mohammad-Reza Malekpour, Naser Ahmadi.

**Formal analysis:** Ali Golestani.

**Funding acquisition:** Nazila Rezaei.

**Investigation:** Sepehr Khosravi.

**Methodology:** Ali Golestani, Mohammad-Reza Malekpour, Farshad Farzadfar.

**Project administration:** Nazila Rezaei.

**Resources:** Ayyoob Jafari.

**Supervision:** Farshad Farzadfar.

**Validation:** Mohammad-Reza Malekpour, Seyed Mohammad-Navid Ataei, Sepehr Khosravi, Saeid Shahraz, Farshad Farzadfar.

**Visualization:** Ali Golestani.

**Writing – original draft:** Ali Golestani, Nazila Rezaei.

**Writing – review & editing:** Ali Golestani, Nazila Rezaei, Mohammad-Reza Malekpour, Seyed Mohammad-Navid Ataei, Sepehr Khosravi, Ayyoob Jafari, Saeid Shahraz, Farshad Farzadfar.

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
