## [Decision Letter · Decision Letter 0]

Dear Dr. Farzadfar,

**As the editor, I would like to offer the following general comments:**

**Introduction:** Please ensure that your introduction is scientifically sound. Start with a general overview and then narrow down to specific details. Incorporate existing solutions to the problem, highlight the research gap, and reference previous studies.

**Language Editing:** I recommend enhancing the overall quality of the English language in your manuscript.

**Abstract:** Please revise your abstract to make it more compelling and ensure it meets the journal's standards. It should serve as a concise summary of the key sections of your manuscript.

Please submit your revised manuscript by Mar 02 2025 11:59PM. If you will need more time than this to complete your revisions, please reply to this message or contact the journal office at plosone@plos.org . Please include the following items when submitting your revised manuscript:

We look forward to receiving your revised manuscript.

Kind regards,

Habtamu Setegn Ngusie

Academic Editor

PLOS ONE

Journal Requirements:

“This work was supported by the National Institute for Medical Research Development (NIMAD), Tehran, Iran [grant number: 940567]”

“The authors gratefully acknowledge that this research benefited from funding from the National Institute for Medical Research Development (NIMAD), Tehran, Iran by grant number 940567. We also thank Kavi Bhalla, PhD, for his contribution to the revision of the manuscript. “

“This work was supported by the National Institute for Medical Research Development (NIMAD), Tehran, Iran [grant number: 940567]”

5. In the online submission form, you indicated that [Insert text from online submission form here].

Reviewers' comments:

**Comments to the Author**

1. Is the manuscript technically sound, and do the data support the conclusions?

Reviewer #1: Yes

Reviewer #2: Partly

2. Has the statistical analysis been performed appropriately and rigorously?

Reviewer #1: Yes

Reviewer #2: Yes

3. Have the authors made all data underlying the findings in their manuscript fully available?

Reviewer #1: Yes

Reviewer #2: No

4. Is the manuscript presented in an intelligible fashion and written in standard English?

Reviewer #1: Yes

Reviewer #2: Yes

Reviewer #1: Strengths:

Comprehensive Dataset: The study leverages a large and rich dataset from 1673 intercity buses integrated with weather and location data, enabling a more holistic view of the factors influencing errors in road accident hotspots.

Multiple ML Models Compared: By examining six different machine learning models (including XGBoost, random forest, and SVM), the study robustly identifies the best-performing method rather than relying on a single algorithm.

Interpretability with SHAP: The use of SHapley Additive exPlanation (SHAP) values offers insight into the most influential predictors, enhancing the understandability of the model’s results and supporting actionable policy decisions.

Attention to Class Imbalance: The study acknowledges the severe imbalance in the outcome variable and utilizes an ensemble approach, increasing the reliability of its predictive performance and reducing bias against the minority class.

Inclusion of Diverse Predictors: Incorporating spatial, temporal, behavioral (fatigue), and weather-related variables provides a multifaceted perspective on road safety issues and highlights the potential interplay between different predictive factors.

Weaknesses:

Lack of Longitudinal Data: The analysis is limited to a single year (2020). Without longitudinal data spanning multiple years, it’s harder to assess trends, seasonality, or long-term changes in predictors.

Domain-Specific Constraints: The generalizability to other regions or transport systems may be limited, as the findings are based on Iranian intercity buses and roads, which may have unique infrastructural, regulatory, or behavioral characteristics.

Data Quality and Completeness: Although preprocessing steps were taken, potential errors or biases in telematics and weather data collection remain. The study relies on the assumption that all variables were accurately measured and recorded.

Limited Exploration of Temporal Impact: While the study includes temporal variables, it notes a limited impact on predictions. Additional, more nuanced temporal analyses (e.g., time-of-day patterns, seasonal variations) might reveal subtler relationships not captured in the current approach.

Potential Unaccounted Predictors: The study may not incorporate all relevant factors, such as detailed driver profiles, road maintenance levels, enforcement measures, or socio-economic attributes of different regions, which could also influence error risk at hotspots.

Use some or all of these papers to help with weakness

Huang, A. A., & Huang, S. Y. (2023). Increasing transparency in machine learning through bootstrap simulation and Shapley additive explanations. PLoS One, 18(2), e0281922.

Fawcett, L., Thorpe, N., Matthews, J., & Kremer, K. (2017). A novel Bayesian hierarchical model for road safety hotspot prediction. Accident Analysis & Prevention, Elsevier.

Islam, M. K., Reza, I., Gazder, U., Akter, R., & Arifuzzaman, M. (2022). Predicting road crash severity using classifier models and crash hotspots. Applied Sciences, MDPI.

Atumo, E. A., Fang, T., & Jiang, X. (2022). Spatial statistics and random forest approaches for traffic crash hot spot identification and prediction. International Journal of Injury Control and Safety, [Volume/Issue pending], Taylor & Francis.

Reviewer #2: Thank you for inviting me to review this interesting scientific paper entitled "Predicting Errors in Accident Hotspots and Investigating Spatiotemporal, Weather, and Behavioral Factors Using Interpretable Machine Learning: an Analysis of Telematics Big Data".

The paper describes about the investigation of big telematics data to predict errors in Accidents Hotspots.

The study utilized modern and sound methods of prediction of outcomes such as machine learning models. I found the paper is sound and would be more sound if the authors consider the following suggestion.

Abstract:

Introduction: The abstract for this manuscript is not clear. Please briefly introduce the outcome and the main gaps you identified.

Methods: the authors identified six machine learning models to predict the RTA. My question is what are the unique criteria to select these models as there are many more models which can aid to predict the RTA? Why you select the six models indicated in the manuscript?

Key words: add the study area, Iran .

Introduction: I found the introduction was sound.

Lines 81-82 (Notably, spatial variations in road traffic injuries demonstrate non-random cluster formations, indicating specific locations as more accident-prone)... Can you mentions the areas where the road traffic accident is more clustered ...this could enable the reader to easily understand which areas are more affected by RTA. also the decision makers could spot it out for better interventions globally or locally.

-lines 86-87...mention the different variables used to predict the behavior's of drivers (such as....)

Methods:

- please provide the geographical map of the study area and locate the spatial distribution of the RTA that has common occurrence using GPS points. That can affirm the telematics data has value for identifying the cluster locations for your outcome variable.

-line 202--remove variable occurring second time.

-lines 203-204...your statement indicated ...we restricted our analysis to data without any missing values, resulting in a final dataset comprising 619,988 rows.... My question is how do you manage missing data as if you could mentioned the methods you employed in managing missing data??

-Train /test splitting for the data

-what was your reason to select the 70 % training and 30 % testing data set? As there are also 80 %and 20 % training and testing classifications. Provide valid reference for this classification.

Result:

-It is not clear that whether the authors employed missing values for which variable. What is the methodological implication for performing missing values? Table 2 indicated the reported errors as a missing value.

-Discussions:

In the discussion section the authors tried to compare the findings with other studies conducted previously reporting using different and best performing model. Does it sound comparing different models while reporting different performances. XGBoost is one model may be best and RF is one for the other. comparing these two models is not sound. The methods used to derive the results is also different.

Minor comments

I recommend the authors to point out the strength and limitation of the study, especially in selecting the machine learning models.

-

References

-consider reference numbers 29,30, 42, 43,46,47

**Do you want your identity to be public for this peer review?** For information about this choice, including consent withdrawal, please see our Privacy Policy

Reviewer #1: No

Reviewer #2: No

---

## [Author Response · Author response to Decision Letter 1]

11 Mar 2025

Response to Editor:

Editor:

As the editor, I would like to offer the following general comments:

Introduction: Please ensure that your introduction is scientifically sound. Start with a general overview and then narrow down to specific details. Incorporate existing solutions to the problem, highlight the research gap, and reference previous studies.

Language Editing: I recommend enhancing the overall quality of the English language in your manuscript.

Abstract: Please revise your abstract to make it more compelling and ensure it meets the journal's standards. It should serve as a concise summary of the key sections of your manuscript.

Authors:

The authors would like to sincerely thank you for your time and thoughtful consideration of our manuscript. We have carefully revised the manuscript to address your and the reviewers’ comments to the best of our ability. We have modified the Introduction section to improve its structure and flow. We first highlighted the importance of RTAs, then discussed key contributing factors, followed by the application of machine learning techniques, the potential of telematics data, and, finally, the aim of our study. Regarding language editing, we have thoroughly reviewed the manuscript and improved its clarity and readability. Additionally, we have revised the Abstract to emphasize the key findings of our study and ensure it aligns with the journal’s standards.

We also adressed journal requirements.

Response to Reviewer 1:

Reviewer comment:

Strengths:

Comprehensive Dataset: The study leverages a large and rich dataset from 1673 intercity buses integrated with weather and location data, enabling a more holistic view of the factors influencing errors in road accident hotspots.

Multiple ML Models Compared: By examining six different machine learning models (including XGBoost, random forest and SVM), the study robustly identifies the best-performing method rather than relying on a single algorithm.

Interpretability with SHAP: The use of SHapley Additive exPlanation (SHAP) values offers insight into the most influential predictors, enhancing the understandability of the model’s results and supporting actionable policy decisions.

Attention to Class Imbalance: The study acknowledges the severe imbalance in the outcome variable and utilizes an ensemble approach, increasing the reliability of its predictive performance and reducing bias against the minority class.

Inclusion of Diverse Predictors: Incorporating spatial, temporal, behavioral (fatigue), and weather-related variables provides a multifaceted perspective on road safety issues and highlights the potential interplay between different predictive factors.

Weaknesses:

Lack of Longitudinal Data: The analysis is limited to a single year (2020). Without longitudinal data spanning multiple years, it’s harder to assess trends, seasonality, or long-term changes in predictors.

Domain-Specific Constraints: The generalizability to other regions or transport systems may be limited, as the findings are based on Iranian intercity buses and roads, which may have unique infrastructural, regulatory, or behavioral characteristics.

Data Quality and Completeness: Although preprocessing steps were taken, potential errors or biases in telematics and weather data collection remain. The study relies on the assumption that all variables were accurately measured and recorded.

Limited Exploration of Temporal Impact: While the study includes temporal variables, it notes a limited impact on predictions. Additional, more nuanced temporal analyses (e.g., time-of-day patterns, seasonal variations) might reveal subtler relationships not captured in the current approach.

Potential Unaccounted Predictors: The study may not incorporate all relevant factors, such as detailed driver profiles, road maintenance levels, enforcement measures, or socio-economic attributes of different regions, which could also influence error risk at hotspots.

Use some or all of these papers to help with weakness

Huang, A. A., & Huang, S. Y. (2023). Increasing transparency in machine learning through bootstrap simulation and Shapley additive explanations. PLoS One, 18(2), e0281922.

Fawcett, L., Thorpe, N., Matthews, J., & Kremer, K. (2017). A novel Bayesian hierarchical model for road safety hotspot prediction. Accident Analysis & Prevention, Elsevier.

Islam, M. K., Reza, I., Gazder, U., Akter, R., & Arifuzzaman, M. (2022). Predicting road crash severity using classifier models and crash hotspots. Applied Sciences, MDPI.

Atumo, E. A., Fang, T., & Jiang, X. (2022). Spatial statistics and random forest approaches for traffic crash hot spot identification and prediction. International Journal of Injury Control and Safety, [Volume/Issue pending], Taylor & Francis.

Authors:

The authors sincerely appreciate the time and thoughtful consideration you have dedicated to reviewing our manuscript. We have modified the "Limitations and Strengths" section of our discussion based on the points you raised. Additionally, we have incorporated the references you suggested to better address the limitations of our work and to deepen the discussion. Regarding the "Limited Exploration of Temporal Impact," we would like to clarify that we did explore the temporal effects on our outcome. However, based on our feature importance analysis, temporal variables were not found to be significant, which is why they were not included in the subsequent ML analysis. Once again, we sincerely thank you for your meticulous comments, which we believe have significantly improved the quality of our work.

Response to Reviewer 2:

Reviewer comment:

Thank you for inviting me to review this interesting scientific paper entitled "Predicting Errors in Accident Hotspots and Investigating Spatiotemporal, Weather, and Behavioral Factors Using Interpretable Machine Learning: an Analysis of Telematics Big Data". The paper describes about the investigation of big telematics data to predict errors in Accidents Hotspots. The study utilized modern and sound methods of prediction of outcomes such as machine learning models. I found the paper is sound and would be more sound if the authors consider the following suggestion.

Authors:

The authors would like to express their most sincere words of appreciation for the time and kind consideration of the reviewer. Thank you for your thoughtful comments, which we believe have significantly improved the quality of our work.

Abstract:

Introduction: The abstract for this manuscript is not clear. Please briefly introduce the outcome and the main gaps you identified.

Thank you for your meticulous comment. We have revised the abstract to better highlight the background, significance, and outcome of our study. Additionally, we have provided more detailed results in the abstract section.

Methods: the authors identified six machine learning models to predict the RTA. My question is what are the unique criteria to select these models as there are many more models which can aid to predict the RTA? Why you select the six models indicated in the manuscript?

Thank you for your thoughtful comment. In many ML studies across different fields, multiple models are tested to identify the best-performing one before conducting further analysis (1-3). There are no strict or mandatory criteria for selecting specific models. For this study, we chose machine learning models commonly used in similar research, ranging from simple and conventional models like logistic regression to more advanced models like XGBoost. We have also highlighted this as one of the strengths of our study (as noted by Reviewer 1) in the discussion section.

Key words: add the study area, Iran .

Thank you for thhoughful comment. Iran added as one of the keywords.

Introduction: I found the introduction was sound.

Lines 81-82 (Notably, spatial variations in road traffic injuries demonstrate non-random cluster formations, indicating specific locations as more accident-prone)... Can you mentions the areas where the road traffic accident is more clustered ...this could enable the reader to easily understand which areas are more affected by RTA. also the decision makers could spot it out for better interventions globally or locally.

Thanks for you meticolus comment. We used the reference of the sentecne and clarified it: Notably, spatial variations in road traffic injuries demonstrate non-random cluster formations, indicating specific locations as more accident-prone like locations with higher traffic interactions and urban areas

-lines 86-87...mention the different variables used to predict the behavior's of drivers (such as....)

Thanks you for your meticolus comment. We used the reference of the sentecne and clarified it: To better understand and predict driving errors leading to RTAs, researchers have employed various techniques incorporating factors such as vehicle speed, acceleration patterns, braking intensity, steering movements, road conditions, and environmental factors like weather and traffic density

Methods:

- please provide the geographical map of the study area and locate the spatial distribution of the RTA that has common occurrence using GPS points. That can affirm the telematics data has value for identifying the cluster locations for your outcome variable.

Thank you for your constructive comment. We have added a map of Iran (as Figure 2) showing the distribution of errors recorded by telematics and the distribution of hotspots. We believe this visualization will help readers better understand the importance of telematics data in identifying error occurrences in accident hotspots.

-line 202--remove variable occurring second time.

Thanks for your accuracy. Addressed.

-lines 203-204...your statement indicated ...we restricted our analysis to data without any missing values, resulting in a final dataset comprising 619,988 rows.... My question is how do you manage missing data as if you could mentioned the methods you employed in managing missing data??

Thank you for your meticulous comment. Please note that we used three datasets in this study, as explained in the methodology. The first dataset was collected via telematics and included variables such as the type of errors, the geographical location of errors (longitude and latitude), the precise timing of errors (including date, hour, minute, and second), and road type, comprising approximately 1.59 million records. The second dataset contained information on 1,492 accident hotspot locations. We calculated the distance between errors recorded by telematics and all hotspot locations. If an error occurred within a 150-meter radius of an accident hotspot, it was considered to have happened in an accident hotspot. The third dataset consisted of weather-related variables obtained from Meteostat using the locations and exact timing of errors. However, complete weather data was not available, and only 690,000 errors had at least one non-missing weather-related variable. Since the highest number of missing values was observed in the weather conditions variable (classified as clear, cloudy, foggy, rainy, snowy, or stormy), we used other available data to impute missing values using the HistGBDT method. The final dataset, which integrated all these variables and had no missing values, consisted of approximately 620,000 records—an adequate sample size for performing machine learning (ML) analyses.

-Train /test splitting for the data

-what was your reason to select the 70 % training and 30 % testing data set? As there are also 80 %and 20 % training and testing classifications. Provide valid reference for this classification.

Thank you for your thoughtful comment. Different textbooks and courses suggest various data splitting ratios, typically ranging between 80:20 and 70:30. There is no strict rule for selecting a specific proportion, and as evident from the cited article (1), different studies have used different splitting ratios.In this study, we opted for a 70:30 split due to the high imbalance in the outcome variable. This approach allowed us to retain as many outcome instances as possible for both the training and testing processes, ensuring better model performance and generalization.

Result:

-It is not clear that whether the authors employed missing values for which variable. What is the methodological implication for performing missing values? Table 2 indicated the reported errors as a missing value.

Thank you for your meticulous comment. The descriptive part of the results section (Overview) is based on the available data. The percentages related to spatiotemporal distribution and locations were reported for 1,583,811 errors, as mentioned in the results section: 'After the initial cleaning of the primary telematics dataset, a total of 1,583,811 errors were available for analysis.'. In Table 2, we provided information on weather-related variables before performing HistGBDT to impute missing values for the 'weather condition' variable. The descriptive results were reported to provide an overview of our dataset. Regarding the outcome variable in our study (errors occurring in accident hotspots), which was complete, 1,277,388 instances were classified as 'no' and 24,894 as 'yes' and had missing values for the 'weather condition' variable. Thus, in Table 2, we did not report errors as missing values.

-Discussions:

In the discussion section the authors tried to compare the findings with other studies conducted previously reporting using different and best performing model. Does it sound comparing different models while reporting different performances. XGBoost is one model may be best and RF is one for the other. comparing these two models is not sound. The methods used to derive the results is also different.

Thank you for your thoughtful comment. We have made some changes in the discussion section to prevent any misunderstanding. Based on our study, both XGBoost and Random Forest, as decision tree-based models, demonstrated the best performance. In the discussion section, we also highlighted that decision tree-based models have performed well in other studies. Additionally, integrating them with SHAP can help address their interpretability challenges. Therefore, we suggest that future studies consider using similar methods to achieve both high accuracy and interpretability.

Minor comments

I recommend the authors to point out the strength and limitation of the study, especially in selecting the machine learning models.

Thank you for your thoughtful comment. Based on your feedback and that of Reviewer 1, we have revised the strengths and limitations section of our discussion to enhance clarity and comprehensiveness.

- References

-consider reference numbers 29,30, 42, 43,46,47

Thank you for your attention to detail. References 29 and 30 correspond to the websites from which we obtained data and information regarding accident hotspots. Accessing these sources requires an Iran-based IP. References 42 and 46 are both well-known textbooks on machine learning. Reference 43 is the original article introducing XGBoost with detailed explanations. Reference 47 was the first article discussing the SHAP method in game theory. However, we have removed it and replaced it with a more recent study that follows a similar methodology to our study.

Once again, we sincerely appreciate your meticulous comments, which we believe have significantly improved the quality of our work. We hope that the revisions we have made, along with our responses, address your concerns and make the manuscript suitable for publication in your view.

Referencess:

1. Silva PB, Andrade M, Ferreira S. Machine learning applied to road safety modeling: A systematic literature review. Journal of traffic and transportation engineering (English edition). 2020;7(6):775-90.

2. Huang AA, Huang SY. Increasing transparency in machine learning through bootstrap simulation and shapely additive explanations. PLoS One. 2023;18(2):e0281922.

3. Iranitalab A, Khattak A. Comparison of four statistical and machine learning methods for crash severity prediction. Accident Analysis & Prevention. 2017;108:27-36.

---

## [Decision Letter · Decision Letter 1]

We look forward to receiving your revised manuscript.

Kind regards,

Habtamu Setegn Ngusie

Academic Editor

PLOS ONE

Journal Requirements:

**Additional Editor Comments:**

The overall quality of the manuscript is good, but it is important to address a few comments before publication, as this journal maintains a high standard quality. This is also vital for the reputation of all experts involved in this manuscript, including the reviewers, the editor, and the esteemed authors.

Please recheck all English copyediting and grammatical issues. For example, in the third sentence of your introduction, change:

"93% of road traffic deaths occurred in low- and middle-income countries (LMIC), despite they have only 60% of vehicles [1]"

to

"About 93% of road traffic deaths occurred in low- and middle-income countries (LMIC), despite having only 60% of vehicles [1]."

Do not start sentences with numerical figures; add something like "about."

In the following sentence of your introduction in paragraph 2, there is no space between "and" and "accident." Review this quoted sentence:

"Various internal and external factors can affect driving, potentially leading to aggressive driving, errors, and accidents."

Please check all grammatical issues from the introduction to the end, as this is vital for your and the journal's reputation.

In the quoted sentence towards the end of your introduction, please remove the phrase "compared to conventional statistical techniques like regressions." The revised sentence should read:

"However, many ML models function as 'black box' systems, lacking interpretability, which limits their application in policymaking [16]."

When referencing your own tables and figures, please bold "Figure 1," "Table 1," and others. For example, when you mention it is further presented in Figure 1, it should be bolded as "Figure 1."

In your Statistical Analyses section, the first sentence states:

"Quantitative variables were described as the mean and standard deviation (SD), while qualitative variables were presented in the form of frequencies and percentages."

My question here is: how can qualitative variables be presented with percentages if they are not quantitative?

I would be very glad if you could incorporate "hyperparameter optimization techniques" and show the ROC curve/AUC before and after tuning or optimization. You can employ only one technique, for example, grid search tuning or Bayesian optimization.

I recommend adding a strengths and limitations of the study section at the end of the discussion and before the conclusion.

Please also highlight headings and subheadings clearly, ensuring the first letter of each heading and subheading is capitalized.

Captions for each figure are essential.

Lastly, the author may benefit from citing the following article for some of their methodological arguments, as it clearly articulates the aspects we should follow in machine learning, especially in predictive modeling; article link: https://link.springer.com/article/10.1186/s12889-024-19566-8

Reviewers' comments:

Reviewer's Responses to Questions

**Comments to the Author**

Reviewer #1: All comments have been addressed

2. Is the manuscript technically sound, and do the data support the conclusions?

Reviewer #1: Yes

3. Has the statistical analysis been performed appropriately and rigorously?

Reviewer #1: Yes

4. Have the authors made all data underlying the findings in their manuscript fully available?

Reviewer #1: Yes

5. Is the manuscript presented in an intelligible fashion and written in standard English?

Reviewer #1: Yes

Reviewer #1: All edits are made satisfactorily. This manuscript has meet the criteria to be accepted by this journal. No further edits needed

**Do you want your identity to be public for this peer review?** For information about this choice, including consent withdrawal, please see our Privacy Policy

Reviewer #1: No

---

## [Author Response · Author response to Decision Letter 2]

17 May 2025

Response to Editor:

Editor:

The overall quality of the manuscript is good, but it is important to address a few comments before publication, as this journal maintains a high standard quality. This is also vital for the reputation of all experts involved in this manuscript, including the reviewers, the editor, and the esteemed authors.

Please recheck all English copyediting and grammatical issues. For example, in the third sentence of your introduction, change:

"93% of road traffic deaths occurred in low- and middle-income countries (LMIC), despite they have only 60% of vehicles [1]"

to

"About 93% of road traffic deaths occurred in low- and middle-income countries (LMIC), despite having only 60% of vehicles [1]."

Do not start sentences with numerical figures; add something like "about."

In the following sentence of your introduction in paragraph 2, there is no space between "and" and "accident." Review this quoted sentence:

"Various internal and external factors can affect driving, potentially leading to aggressive driving, errors, and accidents."

Please check all grammatical issues from the introduction to the end, as this is vital for your and the journal's reputation.

In the quoted sentence towards the end of your introduction, please remove the phrase "compared to conventional statistical techniques like regressions." The revised sentence should read:

"However, many ML models function as 'black box' systems, lacking interpretability, which limits their application in policymaking [16]."

Authors:

The authors sincerely thank you for your time, thoughtful review of our manuscript, and constructive feedback. We have thoroughly revised the manuscript, with particular attention to the grammatical issues you highlighted. We believe that the revised version now meets the standards of scientific writing and is suitable for publication. All changes have been clearly marked using track changes.

Editor:

When referencing your own tables and figures, please bold "Figure 1," "Table 1," and others. For example, when you mention it is further presented in Figure 1, it should be bolded as "Figure 1."

Authors:

Thank you for your feedback. We have bolded the names of figures and tables throughout the manuscript where they are referenced. We have ensured full alignment with the PLOS ONE submission guidelines (as outlined in https://journals.plos.org/plosone/s/file?id=9cba/PLOS%20Manuscript%20Body%20Formatting%20Guidelines.pdf). Specifically, we have referred to tables and figures as Table 1, Fig 1, S1 Table, S1 Fig, etc., in accordance with to PLOS ONE guideline.

Editor:

In your Statistical Analyses section, the first sentence states:

"Quantitative variables were described as the mean and standard deviation (SD), while qualitative variables were presented in the form of frequencies and percentages."

My question here is: how can qualitative variables be presented with percentages if they are not quantitative?

Authors:

Thank you for your attention to detail. The percentages for qualitative (categorical) variables were calculated by dividing the number of cases in a specific category by the total number of cases for that variable. For example, in Table 2, the percentage of error occurrences in accident hotspots under clear weather conditions is 1.86%. This was calculated by dividing the number of errors in accident hotspots under clear weather (3,570) by the total number of errors under clear weather conditions (191,570).

Editor:

I would be very glad if you could incorporate "hyperparameter optimization techniques" and show the ROC curve/AUC before and after tuning or optimization. You can employ only one technique, for example, grid search tuning or Bayesian optimization.

Authors:

Thank you for your insightful suggestion. We have performed grid search for hyperparameter tuning in last version of submission. For this version, We used the citation https://link.springer.com/article/10.1186/s12889-024-19566-8 to show the reason we choosed this approach in the “Machine learning models and evaluation” subsection of the method by:

“While various approaches such as grid search, random search, and Bayesian optimization exist for model optimization, grid search has demonstrated comparable performance. Therefore, it was selected for hyperparameter tuning in this study. Grid search was conducted using stratified 5-fold cross-validation with 3 repeats.”

Although we had previously presented the results of hyperparameter tuning in S4 Table, we have now added the complete results for hyperparameter tuning of all six models in a new supplementary file (S5 Table).

Editor:

I recommend adding a strengths and limitations of the study section at the end of the discussion and before the conclusion.

Authors:

Thank you for your recommendation. It is added.

Editor:

Please also highlight headings and subheadings clearly, ensuring the first letter of each heading and subheading is capitalized.

Captions for each figure are essential.

Authors:

Thank you for your recommendation. We reviewed all headings and subheadings and have applied the appropriate formatting in accordance with the PLOS ONE guidelines. Additionally, figure captions have been provided in the manuscript following the journal's instructions, which recommend including figure legends in the text after their first mention.

Editor:

Lastly, the author may benefit from citing the following article for some of their methodological arguments, as it clearly articulates the aspects we should follow in machine learning, especially in predictive modeling; article link: https://link.springer.com/article/10.1186/s12889-024-19566-8

Authors:

Thank you for your recommendation. The referenced study is indeed valuable, particularly due to its comprehensive and detailed methodology. We have cited this reference as number 45 in our manuscript, in the section where we discuss the rationale for selecting grid search for model optimization.

---

## [Editor Report · Decision Letter 2]

Predicting Errors in Accident Hotspots and Investigating Spatiotemporal, Weather, and Behavioral Factors Using Interpretable Machine Learning: an Analysis of Telematics Big Data

PONE-D-24-49118R2

Dear Dr. Farshad Farzadfar,

We’re pleased to inform you that your manuscript has been judged scientifically suitable for publication and will be formally accepted for publication once it meets all outstanding technical requirements.

Kind regards,

Habtamu Setegn Ngusie

Academic Editor

PLOS ONE

Additional Editor Comments (optional):

Accepted, but the academic editor is flexible if the author makes any editorial changes during the proof stage. The author is also advised to review the proof carefully.
---

## [Editor Report · Acceptance letter]

PONE-D-24-49118R2

PLOS ONE

Dear Dr. Farzadfar,

I'm pleased to inform you that your manuscript has been deemed suitable for publication in PLOS ONE. Congratulations! Your manuscript is now being handed over to our production team.

Kind regards,

on behalf of

Dr. Habtamu Setegn Ngusie

Academic Editor

PLOS ONE